# Competition for Nitrogen Resources: An Explanation of the Effects of a Bioprotective Strain *Metschnikowia pulcherrima* on the Growth of *Hanseniaspora* Genus in Oenology

**DOI:** 10.3390/foods13050724

**Published:** 2024-02-27

**Authors:** Maëlys Puyo, Léa Scalabrino, Rémy Romanet, Scott Simonin, Géraldine Klein, Hervé Alexandre, Raphaëlle Tourdot-Maréchal

**Affiliations:** 1Université Bourgogne Franche-Comté, Institut Agro, Université Bourgogne, INRAE, UMR PAM 1517, 21000 Dijon, France; maelys.puyo@gmail.com (M.P.); geraldine.klein@u-bourgogne.fr (G.K.); rvalex@u-bourgogne.fr (H.A.); 2DIVVA (Développement Innovation Vigne Vin Aliments) Platform, UMR Procédés Alimentaires et Microbiologiques, IUVV, 2 Rue 11 Claude Ladrey, 21000 Dijon, France; remy.romanet@sayens.fr; 3Changins, Viticulture and Enology, HES-SO University of Applied Sciences and Arts Western Switzerland, Route de Duillier 50, 1260 Nyon, Switzerland; scott.simonin@changins.ch

**Keywords:** non-*Saccharomyces*, bioprotection, wine, microbial interactions, *Metschnikowia pulcherrima*, *Hanseniaspora*

## Abstract

As a biological alternative to the antimicrobial action of SO_2_, bioprotection has been proposed to winemakers as a means to limit or prevent grape musts microbial alteration. Competition for nitrogenous nutrients and for oxygen are often cited as potential explanations for the effectiveness of bioprotection. This study analyses the effect of a bioprotective *M. pulcherrima* strain on the growth of one *H. valbyensis* strain and one *H. uvarum* strain. Bioprotection efficiency was observed only against *H. valbyensis* inoculated at the two lowest concentrations. These results indicate a potential species-dependent efficiency of the bioprotective strain and a strong impact of the initial ratio between bioprotective and apiculate yeasts. The analysis of the consumption of nitrogen compounds revealed that leucine, isoleucine, lysine and tryptophan were consumed preferentially by all three strains. The weaker assimilation percentages of these amino acids observed in *H. valbyensis* at 24 h growth suggest competition with *M. pulcherrima* that could negatively affects the growth of the apiculate yeast in co-cultures. The slowest rate of O_2_ consumption of *H. valbyensis* strain, in comparison with *M. pulcherrima*, was probably not involved in the bioprotective effect. Non-targeted metabolomic analyses of *M. pulcherrima* and *H. valbyensis* co-culture indicate that the interaction between both strains particularly impact lysin and tryptophan metabolisms.

## 1. Introduction

In must, highly diverse microorganisms coexist and interact with each other and with the environment. At the beginning of the winemaking process, non-*Saccharomyces* (NS) yeasts are predominant in must and continue to grow until *Saccharomyces cerevisiae* takes over to carry out alcoholic fermentation [1,2,3]. During the early stage of the process, NS populations evolve, mainly in favor of *Hanseniaspora* yeast genera, which are frequently reported as having a negative impact on the organoleptic properties of wines [4,5,6]. In order to avoid economic losses due to microbial spoilage, winemakers traditionally add SO_2_ from the beginning of winemaking and at different key steps of the process [7], but consumers are increasingly demanding chemical-free products. In order to decrease SO_2_ addition during pre-fermentative steps and alcoholic fermentation, the use of selected NS yeasts at harvest or after pressing is recommended to protect must against the development of indigenous yeasts which could lead to a decrease in wine organoleptic properties. In brief, this is the bioprotection strategy. The inoculation of bioprotective strains at early steps of winemaking is aimed at colonizing the environment and establishing interactions between the bioprotective strain and native flora, thus inducing the inhibition of indigenous yeasts. 

A wide range of commercial active dry NS yeasts are currently sold by manufacturers as bioprotective yeasts, mainly from *Metschnikowia* and *Torulaspora* genera [8]. The efficiency of bioprotection was previously proven in field trials with a reduction in indigenous yeasts growth through the addition of bioprotection [9,10,11,12,13,14], without a thorough investigation of the physiological mechanisms involved. Numerous interactions occur in the microbial world and fall into two main groups: direct (cell–cell contact) or indirect interactions (nutrient competition, toxic compound production, quorum-sensing, enzymatic activity, etc.) [15,16]. Studies on the interaction between oenological yeasts have shown that many of these interactions possibly occur in wine. Both *S. cerevisiae* and NS yeasts (including *Metschnikowia* and *Torulaspora*, genera which are used in bioprotection strategy) are able to secrete killer toxins [17,18,19,20,21], as well as other toxic compounds such as pulcherriminic acid for *Metschnikowia* genera [22,23,24,25,26]. Pulcherriminic acid is able to chelate iron in the medium, which leads to the production of a red pigment named pulcherrimin [27,28]. This iron starvation of the medium could lead to an antimicrobial effect. Its efficiency in reducing fungi growth on fruits has already been proved [29,30,31]. Quorum sensing has also been investigated in yeasts, especially in *C. albicans* and *S. cerevisiae*. In these two yeasts, quorum sensing seems to be induced by higher alcohols production, controlling the phenotypic changes in the yeasts, such as filamentous forms or entry in the stationary phase [32,33,34,35,36]. Other NS yeasts, as *T. delbrueckii* or *M. pulcherrima*, can produce higher alcohols [37,38], but the link with putative QS mechanisms have not yet been proven.

However, one of the main hypotheses regarding the implantation and antagonistic effect of bioprotection in winemaking conditions is nutrient competition, especially for nitrogen resources. Nitrogen is one of the most important nutrients in must and a lack of nitrogen resources can lead to sluggish or incomplete fermentation due to limited yeast growth [39,40,41]. Furthermore, co-cultures associating different *S. cerevisiae* strains in medium with limited initial concentrations of nitrogen compared to the control showed that the ability of some strains to compete with others, depending on initial nitrogen availability [42]. Nitrogen resources can be classified into two classes: preferential (nitrogen resources consumed in priority) and non-preferential (resources consumed when concentrations of preferential resources are limited). Not all yeasts share the same preferential resources. Nitrogen resources for non-*Saccharomyces* yeasts are still poorly known, but some studies have already demonstrated that several differences exist between species. For example, a comparative study between five NS species and *S. cerevisiae* showed different consumption kinetics between strains with *S. cerevisiae* which consumed all nitrogen resources after 48 h of growth, likewise for *T. delbrueckii* and *Lachancea thermotolerans* yeasts [43], while for the other NS yeasts studied not all nitrogen resources were consumed over 216 h of growth. Furthermore, amino acids were not consumed in the same order by all the species studied. For example, lysine was among the amino acids consumed first by *M. pulcherrima* while *L. thermotolerans* consumed other nutrients before lysine. In addition, growth conditions (media, temperature, initial nitrogen concentration, etc.) can influence the order and level of amino acid consumption [43,44,45,46].

In oenology, the nitrogen requirement of yeasts has been studied only for about a dozen of NS species and mainly in *S. cerevisiae*/NS interaction studies [44,45,47] but rarely for NS/NS interaction in a bioprotection context [48]. Up to now, interactions occurring between bioprotective yeasts and indigenous flora have not been elucidated and a major gap in knowledge represents a barrier to understanding the mechanisms employed by bioprotection. 

Today, feedback from winemakers on the use of bioprotection shows that the effects of NS addition in grape must can be unpredictable. In order to limit economic losses and control all of the winemaking process, a better management of bioprotection is now essential in strategies aimed at reducing the use of sulfites. Understanding the mechanisms that ensure the effectiveness of bioprotection in oenology is inextricably linked to improving the protocols recommended to winemakers. The purpose of this study was to investigate the antagonistic efficiency of a *M. pulcherrima* bioprotective yeast against one strain of *Hanseniaspora uvarum* and one strain of *Hanseniaspora valbyensis* at three initial concentration levels in laboratory conditions on synthetic must. The bioprotective effect was determined using growth monitoring in single and co-cultures. In order to elucidate interaction between the bioprotective and apiculate yeasts at the pre-fermentative stage, their nitrogen and oxygen requirements were studied. A metabolic footprint of the interaction was also investigated through un-targeted metabolomic analyses.

## 2. Materials and Methods

### 2.1. Strains

The yeast *Metschnikowia pulcherrima* MCR24 (Primaflora VB^®^—AEB Group, Kaysersberg-Vignoble, France) was used as bioprotective yeast. Two apiculate yeasts were used: one strain of *Hanseniaspora uvarum* (strain 3137) and one strain of *Hanseniaspora valbyensis* (strain ScS), isolated from grape musts. All the strains were stored in glycerol 20% (*v*/*v*) at −80 °C.

### 2.2. Media and Growth Condition

Each strain was spread before use on YPD medium (glucose 20 g/L, yeast extract 5 g/L, tryptone 10 g/L, chloramphenicol 0.2 g/L, agar 16 g/L) and incubated at 28 °C for 48 h. A two-step preculture was prepared for each strain. The first preculture was prepared from a single colony taken from a YPD plate and incubated in YPD liquid medium for 24 h at 28 °C under stirring at 130 rpm. The second preculture was prepared with 50% of the first preculture and 50% of fresh YPD liquid medium and incubated in the same conditions as the first preculture.

Cultures were performed in a conic plastic TubeSpin^®^ Bioreactor 50 (TPP Techno Plastic Products AG, Zollstrasse 7, Trasadingen, Switzerland) with an air-permeable stopper while preserving the sterility of the samples. Tubes were filled with 40 mL of synthetic must (MS300 containing 300 mg N/L (adapted from Bely et al. [39] and described in Evers et al. [49] (Appendix A)). A single culture of each strain was grown with an initial concentration of 5 × 10^5^ CFU/mL for *M. pulcherrima* MCR24 (initial concentration recommended by manufacturer), and at three different initial concentrations for both *Hanseniaspora* strains, 5 × 10^4^, 5 × 10^5^ and 5 × 10^6^ CFU/mL, in order to reproduce three levels of must contamination. Co-cultures with *M. pulcherrima* MCR24 were grown with a single inoculation rate set at 5 × 10^5^ CFU/mL for *M. pulcherrima* MCR24 and each *Hanseniaspora* strain independently at three different initial concentrations, as monocultures. Cultures were conducted for 72 h at 20 °C. A sample of each condition was taken sterilely every 3 h for the first 48 h and every 6 h from 48 h to 72 h. A total of 350 µL was sampled for each time point, 100 µL was used for cell enumeration and the remaining 250 µL left was stored at −20 °C for chemical analyses. Each condition was performed in quadruplicate.

### 2.3. Growth Analysis

Growth kinetics were monitored through colony enumeration on agar plates. The samples were diluted when needed in physiological water. *M. pulcherrima* strain enumeration was performed on WL Oxoïd CM039 medium with the addition of 0.2 g/L of chloramphenicol (Sigma-Aldrich, Saint-Quantin-Fallvier, France) when they presented red colonies due to the production of pulcherrimin [28]. *Hanseniaspora* strains were enumerated on selective ITV medium (20 g/L glucose, 10 g/L yeast extract, 20 g/L tryptone, 0.1 g/L para-coumaric acid, 0.1 g/L ferulic acid, 0.03 g/L green bromocresol, 0.2 g/L chloramphenicol, 20 g/L agar, pH 5, with addition of cycloheximide 0.006% (*v*/*v*) [50]). Colonies were enumerated after agar plate incubation at 28 °C for 48 h. Maximal growth rate (µmax) was calculated with the RStudio Package “Growthrates”.

### 2.4. Nitrogen Analysis

For each single culture biological replicate (four for each condition), amino acid concentration was measured through High Pressure Liquid Chromatography (HPLC), and ammonium content (for single and cocultures) using a manual enzymatic kit (no. 030) (BioSentec, Portet-Sur-Garonne, France).

The HPLC analyzer used for the amino acids was an Agilent 1260 Infinity (Agilent, Santa Carla, CA, USA), with a column Poroshell HPH C18 (100 × 4.6 mm) with 2.6 µm particle size (Agilent, Santa Clara, CA, USA). The analytical method used for amino acid separation was an Agilent method with an automated system for online OPA/FMOC derivatization according to the Agilent application note number 5991–5571EN (https://www.agilent.com/cs/library/applications/5991-5571EN.pdf, accessed on 2 November 2017). The separation methods used two eluants. Mobile phase A was composed of Na_2_HPO_4_ (1.4 g/L) (Sigma-Aldrich, Saint-Quantin-Fallvier, France), 10 mM Na_2_B_4_O_7_.10H_2_O (3.8 g/L) (Sigma-Aldrich, Saint-Quantin-Fallvier, France), pH 8.20, and mobile phase B was a mixture of acetonitrile/methanol/water (45/45/10). The method used a constant flow rate of 2 mL/min, with a gradient provided in Appendix A. The column oven was set at 40 °C. Amino acid auto-derivatization was set up as follows: 2.50 µL of borate buffer (placed in a 2 mL vial in an autosampler) was collected in the sample loop with 0.50 µL of sample and mixed twice in the loop. After waiting for 30 s, 0.50 µL of OPA (placed in a 2 mL vial in an autosampler) was collected and added in the loop and mixed 6 times with the mixture of borate and sample. After waiting for 18 s, 0.50 µL of FMOC (placed in a 2 mL vial in an autosampler) was collected and mixed 6 times. After waiting for 24 s, 32 µL of injection eluant (eluant A + 4 mL/L H_3_PO_4_, placed in a 2 mL vial in an autosampler) was added in the loop and mixed with the mixture of OPA/FMOC/sample, then 0.5 µL of the derivatized sample was injected in the column.

### 2.5. Oxygen Requirement

Dissolved oxygen was monitored in conic plastic tubes with impermeable stoppers containing 40 mL of synthetic must (MS300) previously saturated in O_2_. Medium was inoculated in single culture at the same cell concentrations used previously and incubated at 20 °C. Dissolved oxygen concentration in must was monitored until complete consumption with the Nomasens^®^ P300 analyzer (Vinventions WQS ©, Rivesaltes, France) with Pst3 pads (Vinventions WQS ©, Rivesaltes, France) stuck inside the tubes. 

### 2.6. UHPLC-Q-ToF-MS/MS Untargeted Analysis

The composition of non-volatile compounds was analyzed using ultra-high pressure liquid chromatography (UHPLC) (Dionex Ultimate 3000, ThermoFisher, Waltham, MA, USA) coupled to a MaXis plus MQ ESI-Q-ToF mass spectrometer (MS) (Bruker, Bremen, Germany), as used in Evers et al. [49].

Each sample (single and co-culture in quadruplicate) at the final time point (after 72 h of growth at 20 °C) was centrifuged at 10,500× *g* for 10 min and kept at 10 °C during analysis. Nonpolar compounds were separated with reverse phase liquid chromatography in a UPLC Acquity BEH C18 1.7 µm column 100 × 2.1 mm (Waters, Guyancourt, France), using mobile phase A (5% (*v*/*v*) acetonitrile and 0.1% (*v*/*v*) formic acid) and mobile phase B (acetonitrile with addition of formic acid at 0.1% (*v*/*v*)) with a constant flow rate of 0.4 mL/min and an elution temperature of 40 °C. Compounds separation was completed with the following gradient: 5% (*v*/*v*) solvent B from 0 to 1.10 min, before increasing its proportion from 1.10 to 6.40 min to reach 95%, until the end of the analysis (10 min in all). Sample ionization was performed in both negative and positive mode, with a nebulizer pressure set at 2 bar and a dry nitrogen flow set at 10 L/min. The parameters of the mass spectrometer were set for iron transfer at 500 V, for capillary voltage at 4500 V and 3500 V (for positive and negative ionization mode, respectively), for mass range acquisition between 100 and 1000 *m*/*z*, and for fragmentation at 8 Hz spectra rate using the auto MS/MS function (20–50 eV). Calibration was done with an external calibration of Na formate injected before sample analysis in enhanced quadratic mode (error < 0.5 ppm). Analysis repeatability was checked via regular injection of quality control which corresponded to the sample’s mixture.

For the pre-treatment of masses, Burker Compass MetaboScape Software (v. 8.0.1, Bruker, Mannheim, Germany) with the integrated tools SmartFormula using an isotopic profile (parameters set up at mSigma < 20, and 5 ppm) was used. The masses extracted were those with an intensity higher than the threshold of 1000 and which were found in at least 20% of the samples, with 40% for recursive parameters. Based on their putative elemental formula (with carbon, hydrogen, oxygen, nitrogen and sulfur atoms), features selected using statistical analysis were assigned hypothetical chemical families (peptides, amino sugars, aromatic compounds, lipids, carbohydrates) according to the compound classification of Rivas-Ubach et al. [51]. These features were also compared to an in-house database and the online databases Yeast Metabolome Database (YMDB) and Kyoto Encyclopedia of Genes and Genomes (KEGG) to assign them a putative annotation. The level of confidence for annotation was determined according to Schymansky et al. [52].

### 2.7. Statistical Analysis

Statistical analyses and graphical representation were completed with RStudio Software (version 4.2.2). The comparison between growth parameters, nitrogen and oxygen consumption parameters was analyzed using the Kruskal–Wallis test with the post hoc test of Dunn when needed (α = 5%). For the metabolomic analyses, relative intensity of features was analyzed using a *t*-test (n = 2) or a One-Way ANOVA (n = 3) followed by a Tukey post hoc test when needed (α = 5%).

## 3. Results and Discussion

### 3.1. Effect of Interaction on Growth Kinetics

The growth kinetics of each strain in single and co-cultures at three levels of inoculation with *Hanseniaspora* are presented in Figure 1. Maximal growth rate (µmax) and final population values reached by each strain were extracted. A comparison of these values between single culture and co-culture for each strain was carried out in order to determine the interaction effect on growth for each strain. 

#### 3.1.1. Interaction between *Metschnikowia pulcherrima* MCR24 and *Hanseniaspora uvarum* 3137

The analysis of the maximal growth rate (µmax) values of *H. uvarum* showed that whatever its initial concentration, the µmax value was not significantly impacted by the interaction with *M. pulcherrima* (Table 1a, Figure 1). Concerning *M. pulcherrima*, the µmax value was not impacted by the presence of *H. uvarum* when the apiculate yeast was inoculated at 5 × 10^4^ or 5 × 10^5^ CFU/mL. But the growth rate was negatively impacted in co-culture with *H. uvarum* inoculated at 5 × 10^6^ CFU/mL, with a significant decrease of 27% of the µmax value (Table 1a). As for the µmax values, maximal populations reached after 72 h of growth by *H. uvarum* were similar in single and co-cultures whatever the initial population, in contrary to *M. pulcherrima* which was negatively impacted by the interaction with *H. uvarum* whatever the co-culture condition (Table 1b, Figure 1). We also noticed that the higher the initial concentration of *H. uvarum*, the lower the maximum population reached by *M. pulcherrima*. The maximal biomass was lower by about half a log in the co-culture with the initial inoculum of 5×10^4^ CFU/mL with *H. uvarum* and lower by 1.5 log with 5 × 10^6^ CFU/mL for the initial population of *H. uvarum* (Table 1b).

In our conditions, it appeared that whatever the initial *H. uvarum* concentration, *M. pulcherrima* MCR24 was not able to protect the must by limiting *Hanseniaspora* growth. Indeed, a reverse effect was observed, with a negative impact of the growth of *H. uvarum* on the development of the bioprotective strain. 

#### 3.1.2. Interaction between Metschnikowia pulcherrima MCR24 and Hanseniaspora valbyensis ScS

The comparison of µmax values between single and co-cultures (Table 2) showed that *H. valbyensis* was significantly impacted by the interaction with *M. pulcherrima,* with a drop in µmax values of 35% and 55% when *H. valbyensis* was inoculated at 5 × 10^5^ CFU/mL and 5 × 10^4^ CFU/mL, respectively. For the inoculation rate of 5 × 10^6^ CFU/mL, no effect of co-culture on the µmax value of *H. valbyensis* was observed (Table 2a, Figure 2). Whatever the initial *H. valbyensis* concentration, the maximal population reached was negatively impacted in co-culture with the bioprotectant yeast, with more than 1 log of population reduction for the lowest initial concentration of *H. valbyensis* (Table 2b, Figure 2). In contrast to co-cultures with *H. uvarum*, *M. pulcherrima* was not impacted by the interaction regarding either its µmax or its maximal population values. A significative difference was observed only for its maximal population in the co-culture with 5 × 10^6^ CFU/mL of *H. valbyensis* (Table 2a,b).

The comparative analysis of growth behavior suggests a bioprotective effect of *M. pulcherrima* depending on the species of *Hanseniaspora*. Indeed, its efficiency was demonstrated when inoculated with *H. valbyensis* but was not found when inoculated with *H. uvarum.* These results may seem surprising in view of the previous results from the field trials, in which the bioprotective strain was shown to be established and to predominate over the indigenous flora, even when a high proportion of *H. uvarum* was present [10,53,54]. These results need to be confirmed on other strains belonging to these species. Indeed, in some cases, *H. uvarum* yeasts could be found after alcoholic fermentation even after bioprotection or SO_2_ addition during pre-fermentative steps which could indicate different sensitivities according to winemaking conditions or strains [9], and could be explained by the considerable genetic diversity among *Hanseniaspora* clades [55]. Furthermore, in real conditions a higher number of species and strains coexists, leading to considerable interaction diversity. The bioprotective effect could also be the result of a multitude of interactions between the bioprotective strain and the indigenous microbiota.

In addition, this analysis pointed to the importance of the initial ratio between *M. pulcherrima* and apiculate yeasts. Currently, even in conditions where *M. pulcherrima* exhibited a bioprotective effect, this efficiency was repressed when inoculated at a lower concentration than the apiculate yeasts. During trials carried out in a cellar, the work of Windholtz et al. [12] demonstrated limited implantation and a loss of efficiency of bioprotective strains in conditions with grapes at advanced maturity compared to grapes at technical maturity which have a lower indigenous flora concentration. 

### 3.2. Nitrogen Requirements

The nitrogen requirements of *M. pulcherrima* and both *Hanseniaspora* strains were investigated to determine if a possible competition for nitrogen could explain the bioprotective effect observed previously, especially on *H. valbyensis*. The consumption kinetics of 18 amino acids and ammonium for the three strains are shown in Appendix A. Table 3 summarizes the percentage of each amino acid and ammonium consumed after 72 h of culture at 20 °C. The nitrogen resources were not fully consumed, regardless of the strains and the initial concentration used. Ammonium, cysteine and glycine were not consumed and certain others, such as arginine and histidine, were only partially consumed, at 10% and approx. 30–40% consumption, respectively, by all the strains. 

Both *Hanseniaspora* strains, independently of species, exhibited comparable nitrogen consumption for most of the consumed resources. Aspartic acid, glutamic acid, alanine, phenylalanine, serine and tyrosine were heavily metabolized, while the consumption of the *M. pulcherrima* strain was less than 5%. Methionine was totally consumed by both strains of *Hanseniaspora*, compared with 35% for *M. pulcherrima*. When comparing the two strains of *Hanseniaspora,* these results also highlight a higher consumption of aspartic acid (98%) and serine (90%) for *H. valbyensis*, independent of the inoculation rate of both strains (Table 3). 

Only four amino acids were almost entirely consumed by the three strains (over 70%) in all conditions after 72 h and were the first to be consumed and become limiting in all conditions: isoleucine, leucine, lysine and tryptophan (Appendix A, Appendix A). In an interaction context, these resources could be considered as common preferential resources for all conditions. These results strongly suggest a possible competition for these amino acid resources, consumed preferentially during co-cultures. An analysis of the consumption kinetics for these four amino acids enabled the definition of each strain and for all inoculation conditions using the following parameters: the percentage of amino acid consumed in 24 h (% 24 h), the time before the start of amino acid consumption (Lag), the rate and specific maximal rate of amino acid consumption (Rs and Qs) (Table 4). 

For both *Hanseniaspora* strains, consumption rates (Rs in ng/L/h) increased related to their initial inoculation levels. The higher consumption rates were due to the higher cell number and not to higher specific consumption rate values (Qs) by cells whose values tended to fall (Table 4) in line with the decreases in the specific growth rate values previously observed as a function of the level of inoculum. 

*H. uvarum* was characterized by very short latency times (lag times) before the consumption of the four amino acids (between 5 and 7 h), correlated with the highest amino acid consumption values measured after 24 h of growth (Table 4). Lag times logically decreased as the initial population increased. The immediate consumption of the four amino acids was even observed when the initial inoculum was set at 5 × 10^6^ CFU/mL. For the lowest inoculum of *H. uvarum*, lag times for the assimilation of Ile, Leu and Lys were slightly higher than those measured for *M. pulcherrima*. The lower values for the maximum specific rate of assimilation of these three amino acids (Qs) calculated for the bioprotective strain may explain the absence of any negative impact of *M. pulcherrima* on the growth of *H. uvarum*, or even the negative effect of the 3137 strain on the growth of the bioprotective strain for the highest inoculation rate. 

A very interesting feature of the *H. valbyensis* strain was its different behavior. For the lowest inoculation rate (5×10^4^ CFU/mL), the lag times were longer (between 11 and 7 h), correlated with low percentages of amino acids consumed at 24 h in comparison with *M. pulcherrima*. For the highest inoculation rate, lag time values were of the same order of magnitude as those for *M. pulcherrima* for the four amino acids with comparable percentages of consumption at 24 h, except for lysine which was almost totally consumed by *H. valbyensis* after 24 h (87% vs. 50% by *M. pulcherrima*) (Table 4). The different kinetics of assimilation of isoleucine, leucine, lysine and tryptophan by these two strains clearly suggest that in co-cultures, for the lowest *H. valbyensis* inoculation rates, a depletion in these preferential amino resources had a negative impact on its growth. This phenomenon of competition for preferential nitrogenous resources could partly explain the positive effect of the bioprotective strain observed previously (Figure 1).

Among the amino acids most consumed by *M. pulcherrima*, lysine is the one most often classified in the literature as a preferred amino acid [38,44,46,56]. Our results highlighted that this amino acid was the first one to be depleted in the medium during the growth of the bioprotective strain, in accordance with previous data. Numerous studies also pointed out that branched chain amino acids, including leucine and isoleucine, were found as good or intermediate sources [38,44,46,56,57], but Roca-Mesa et al. (2020) [43] showed that in synthetic must at 22 °C under stirring the *M. pulcherrima* strain studied did not consume isoleucine and only partially consumed leucine even after 216 h growth. Most of those studies, in line with our results, suggest that the preferential consumption of these amino sources is a characteristic of the *M. pulcherrima* species, unlike tryptophan, which is often reported as hardly or not consumed by *M. pulcherrima* in many studies [38,43,46,56], and less frequently as consumed [48,57].

Concerning *H. valbyensis*, to our knowledge, no data are available yet concerning the nitrogen requirement of this species, and few investigations have been reported concerning *H. uvarum*. Roca-Mesa et al. (2020) [43] reported that leucine and isoleucine were among the nitrogen sources most consumed with the rapid depletion of these amino acids (between 12 and 48 h at 22 °C under stirring), and the partial consumption of lysine (around 80% consumed in 72 h) and tryptophan (around 50% consumed in 72 h) [43]. Other studies have also reported that these four amino acids are classified as “good” or “intermediate” for supporting different growth parameters, supporting our results for the two strains of *Hanseniaspora* [48,57]. 

To our knowledge, the nitrogen requirements of NS yeasts have very rarely been studied in the context of investigating the mechanisms involved in bioprotection. Under our experimental conditions, the demonstration of the same requirements in amino acids preferentially assimilated but with different assimilation kinetics between the bioprotective strain and *Hanseniaspora* strains reinforces the strong hypothesis of competition for these resources during co-cultures, explaining the limits of bioprotection efficiency. Although competition for nitrogen resources seems to be linked to the bioprotective effect, other nutrient competitions may also take place and contribute to bioprotective efficiency. Indeed, it has been shown that a vitamin or lipid deficiency in the must can impact the growth of some microorganisms [41,45,58,59,60,61].

### 3.3. Oxygen Requirements

Oxygen was another resource naturally found in must after pressing which could be consumed by yeasts and so could lead to possible competition between microorganisms.

The parameters of oxygen consumption for each condition of growth (monocultures) are presented in Table 5. The maximal consumption rates of O_2_ (expressed in mg/L/h) were obtained with *H. uvarum* inoculated at 5 × 10^6^ CFU/mL with the total consumption of dissolved oxygen (DO) occurring in 2.2 h. A significantly lower rate of consumption was observed at the lowest inoculation rate with this strain, compared with *M. pulcherrima*, as well as a 3.5-fold increase in time for the total consumption of DO. Even for this lowest inoculation rate, no negative effect was observed on its fitness in co-cultures with *M. pulcherrima* (Figure 1). This suggests that the rapid disappearance in the medium of the DO concentration during co-culture due to the presence of the bioprotective strain had no impact on its growth. 

The same conclusions can be drawn concerning *H. valbyensis*. This strain was characterized by very low oxygen consumption rates independently of the inoculation rate, with higher total DO consumption times even at the highest inoculation rate (4.5 h vs. 3 h for *M. pulcherrima*) (Table 5). These results underline the fact that this strain, unlike *H. uvarum*, required little oxygen, and rapid DO depletion due to the presence of *M. pulcherrima* in co-culture had no significant effect on its fitness. The competition for oxygen was therefore probably not responsible for limiting the growth of *H. valbyensis* by *M. pulcherrima*.

In the literature, among the non-*Saccharomyces* species studied, *M. pulcherrima* is found to be one of the most rapid consumers of oxygen in the environment [62]. In co-culture with *S. cerevisiae,* as well as in single culture, oxygen addition improves the viability of non-*Saccharomyces* yeasts, its persistence in the medium and growth [63,64]. This improvement of viability linked to an increase in oxygen in the medium has also been reported for *M. pulcherrima*. The addition of oxygen to a *M. pulcherrima*/*S. cerevisiae* co-culture slowed and delayed the death of *M. pulcherrima* by 1 day [65]. In addition, Quirós et al. [66] showed that *M. pulcherrima* has a respiratory quotient of 1 at the beginning of fermentation, implying that this species consumes sugars via respiration and not through fermentation as long as O_2_ is available, which confirms the considerable consumption of this resource by this species. All the previous studies on the impact of oxygen on the growth of non-*Saccharomyces* yeasts had been conducted in the context of interaction with *S. cerevisiae* and not in a bioprotection context. However, they highlighted the strong impact of O_2_ on the survival and persistence of non-*Saccharomyces* yeasts in the environment, as well as the high consumption of this resource by *M. pulcherrima*. In a bioprotection context, this high oxygen consumption could lead to competition between bioprotective strains and indigenous yeasts. 

For *Hanseniaspora*, few data are available on their oxygen requirement. The study of Visser et al. [67] investigated the capacity of diverse yeasts species to grow in anaerobic conditions. The ability to grow in these conditions was shown by both species, *H. uvarum* and *H. valbyensis*. Furthermore, a more recent work reported the ability of many strains of *H. uvarum* genera as well as other *Hanseniaspora* species to grow in anaerobic conditions [55], confirming our hypothesis that *M. pulcherrima*-induced oxygen depletion has no effect on the interactions observed with *Hanseniaspora*. 

### 3.4. Untargeted Metabolomic Analyses of Non-Volatile Compounds

In order to deepen the investigations into the mechanisms involved in bioprotection, we chose to focus only on the pair *M. pulcherrima*/*H. valbyensis*. An untargeted metabolomic analysis allowed the obtainment of a global and unbiased idea of the chemical composition of the medium at a specific time point. The end points (72 h of growth) of co-cultures at the initial concentrations of 5 × 10^4^ and 5 × 10^6^ CFU/mL, for *H. valbyensis* with *M. pulcherrima* with their associated single cultures, were analyzed. The comparison of the results leads to the establishment of a global metabolomic footprint induced by the interaction between the bioprotective strain and *H. valbyensis.*

According to the 3028 features extracted, the samples (single and co-cultures) were represented on a principal component analysis (PCA) on the first two dimensions (dimension 1 and dimension 2 representing 15.7% and 9.5% of result variability, respectively) (Figure 3). The single culture of *M. pulcherrima* was clearly discriminated from the *H. valbyensis* single culture of both initial concentrations on the first dimension. The co-culture with an initial *H. valbyensis* concentration at 5 × 10^6^ CFU/mL was discriminated from the single culture of *M. pulcherrima* and was found on the same area as the single culture of *H. valbyensis* (at 5 × 10^6^ CFU/mL). For co-cultures at an initial *H. valbyensis* concentration of 5 × 10^4^ CFU/mL, the metabolomic footprint was different from that of the *H. valbyensis* single culture. 

These results show that under conditions where bioprotection had the most pronounced effect (initial concentration of 5 × 10^4^ CFU/mL in *H. valbyensis*), the metabolic footprint of the co-culture was further distanced from that of the single culture of *H. valbyensis*, in contrast to the case of a lack of bioprotective effect (5 × 10^6^ CFU/mL condition). To investigate the impact of the interaction and the mechanisms of bioprotection at the metabolomic scale, we focused on the exo-metabolome of single cultures and co-culture for an initial concentration of 5 × 10^4^ CFU/mL in *H. valbyensis*.

A total of 2844 features were extracted (Figure 4a) for the analyses of *M. pulcherrima* and *H. valbyensis* single cultures and the associated co-culture. Some of them were unique to a specific condition (32, 3 and 2 unique features for *M. pulcherrima, H. valbyensis* and the co-culture of both strains, respectively) and some of them were common to several conditions. Statistical analyses led to the extraction of biomarkers specific to each condition. The biomarkers of a condition are defined as features significantly more intense (*p*-value < 0.05) than in other conditions.

The analysis of the chemical composition of the biomarkers showed that the two chemical families represented most were the CHON and CHONS groups in all conditions, but with different distributions (Figure 4b–d). A total of 141 biomarkers were obtained in single culture of *M. pulcherrima*. They corresponded mainly to CHONS-type compounds which could be associated with sulfur-containing peptides, correlated with the high proportion of biomarkers assigned to putative peptide families. For the *H. valbyensis* single culture (87 biomarkers), CHON and CHONS-type compounds were the most abundant with a large proportion of aromatic compounds. While among the biomarkers of co-culture, CHON-type compounds seemed predominant, mainly associated with aromatic compounds (16 biomarkers). The hierarchical clustering analysis of biomarkers of the co-culture and the two single cultures allowed the discrimination of the three conditions. However, it is interesting to note that even if the *H. valbyensis* population was negatively impacted by the interaction, the co-culture seemed to be closer to the *H. valbyensis* single culture than to the *M. pulcherrima* one (Appendix A). 

The putative annotation of some of the co-culture biomarkers in the KEEG, YMDB and in-lab databases was carried out. Among them we found glutaric acid (Annotation level 3), which corresponds to an intermediate in the metabolic pathway of lysin in bacteria and yeasts (KEEG Pathway sce00310, Han et al. 2020 [68] and Prothstein and Hart 1964 [69]), and it was one of the preferentially assimilated amino acids in both strains. Another biomarker of the co-culture was annotated putatively as indole-3-acetaldehyde (Annotation level 3), which is an intermediate compound in the biosynthesis of tryptophol from tryptophan [36], which is one of the most consumed amino acids. In the literature, tryptophol was documented in *C. albicans* as a quorum sensing molecule inducer. The production of tryptophol was reported in *S. cerevisiae*, with a possible implication in quorum sensing signaling, as well as in non-*Saccharomyces* yeasts including *M. pulcherrima.* More investigations are still needed to elucidate the existence of quorum sensing and the implication of tryptophol in its signaling.

These first results did not give a clear answer to the metabolic changes induced during a negative interaction generated by the bioprotective strain, but biomarker annotation tools open the way to further investigations into the nitrogen metabolism of yeasts as well as the investigation of aromatic compounds (such as tryptophol) produced when *M. pulcherrima* is in co-culture with *H. valbyensis* in a bioprotection strategy.

## 4. Conclusions

The use of bioprotection in oenology now appears to be a real alternative to the addition of a chemical input (SO_2_) to protect grape juice against microbiological alteration at the pre-fermentation alcoholic stages. Numerous field trials have demonstrated the effectiveness of this strategy, although there are no clear data on the physiological mechanisms responsible for this effectiveness. Many studies have highlighted the existence of indirect interaction mechanisms under oenological conditions, including competition with nitrogenous nutrients and/or oxygen [41,47,58,70], but no proof has yet been found during the application of bioprotection on must. To our knowledge, this work is the first which clearly demonstrates that nutritional competition for preferential amino acids can contribute to the effectiveness of bioprotection. Of course, the species-dependent effect must be more investigated. Better understanding of biodiversity within the genus *Hanseniaspora,* such as in the work initiated by Albertin et al. [55] as well as other studies [71,72] will in the future enable us to refine our diagnosis of the effectiveness or otherwise of a bioprotective strain and to better decipher the physiological mechanisms involved. Furthermore, temperature and growth conditions seem to be essential factors to consider. Indeed, recently the impact of nitrogen competition was not demonstrated at 12 °C between *Metschnikowia* strains and *Hanseniaspora* mix species [48]. However, this work confirmed previous results [48], showing the importance of knowing the initial concentration of indigenous yeasts in order to adapt must bioprotection protocols in the future. 

## Figures and Tables

**Figure 1 foods-13-00724-f001:**
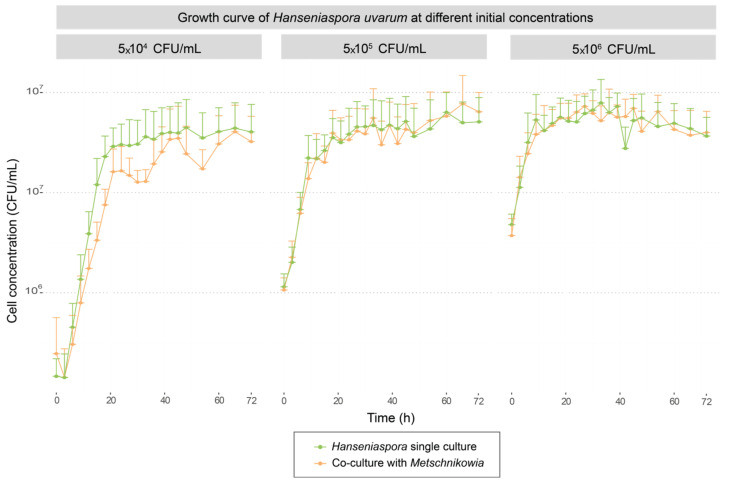
Growths kinetics of *H. uvarum* in single culture (green curve) and in co-culture with *M. pulcherrima* (orange curve) for the three initial concentrations.

**Figure 2 foods-13-00724-f002:**
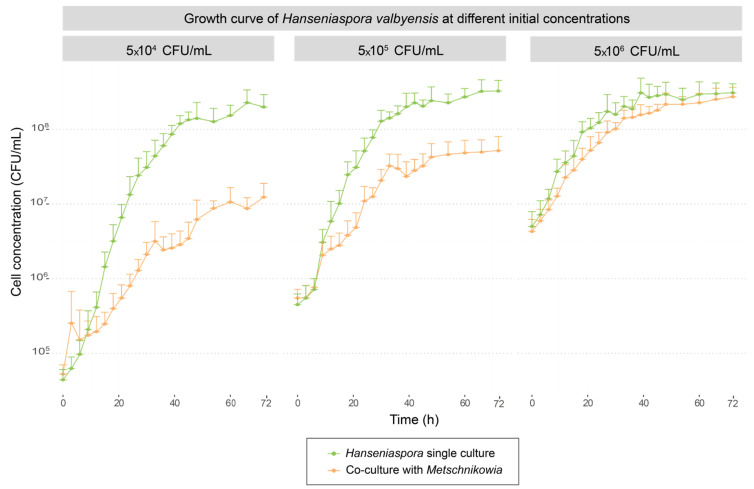
Growths kinetics of *H. valbyensis* in single culture (green curve) and in co-culture with *M. pulcherrima* (orange curve) for the three initial concentrations.

**Figure 3 foods-13-00724-f003:**
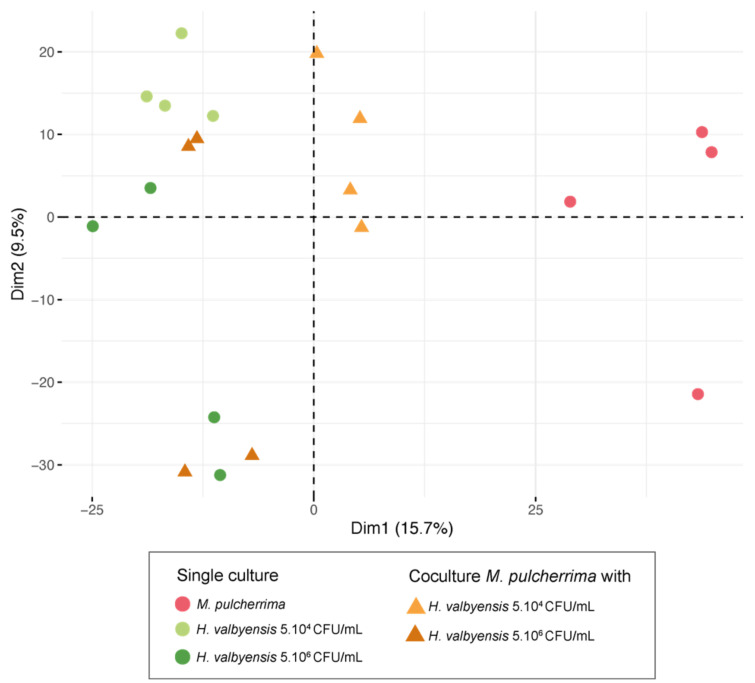
Principal component analyses (PCA) of the single cultures of *M. pulcherrima* and *H. valbyensis* (5 × 10^4^ and 5 × 10^6^ CFU/mL initial concentration) and co-cultures associated according to the features extracted by UHPLC-qToF-MS/MS (3028 features).

**Figure 4 foods-13-00724-f004:**
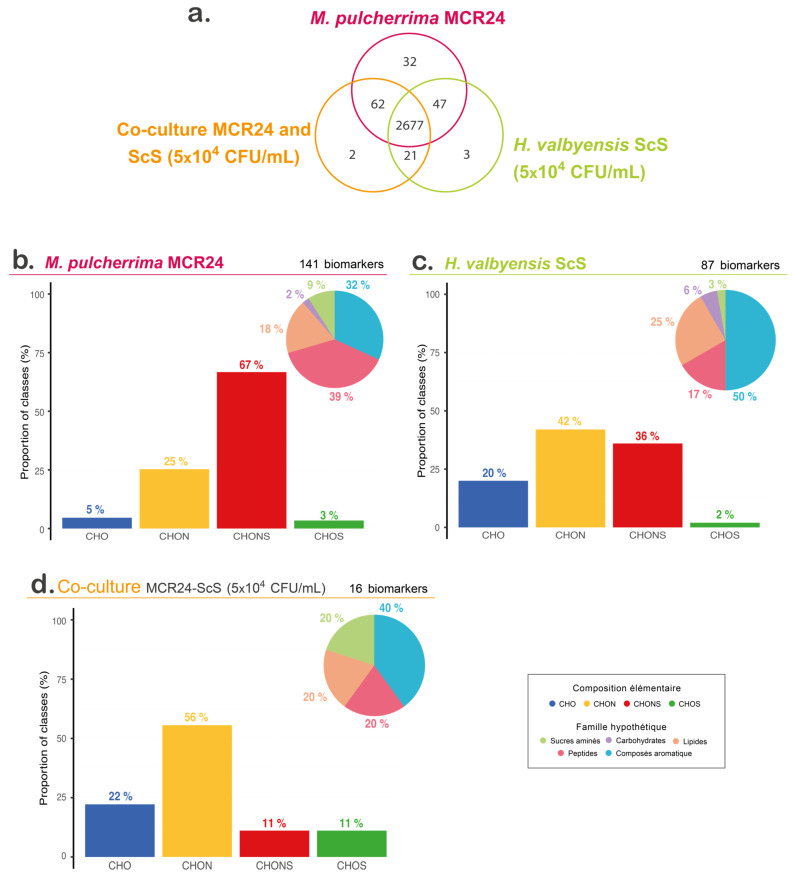
Metabolomic analyses (**a**) of features distribution on a Venn diagram, and biomarkers for (**b**) *M. pulcherrima* single culture, (**c**) *H. valbyensis* (5×10^4^ CFU/mL), (**d**) co-culture of both strains.

**Table 1 foods-13-00724-t001:** Comparison between *H. uvarum* strain 3137 and *M. pulcherrima MCR24* in single and co-cultures for: (**a**) maximal growth rate (µmax (h^−1^)) and (**b**) maximal population (CFU/mL) after 72 h of growth at 20 °C.

(**a**) Table of µmax (h^−1^)				
	***Hanseniaspora uvarum* 3137**		***Metschnikowia pulcherrima* MCR24**	
**Initial Population of *Hu* (CFU/mL)**	**Single Culture**	**Co-Culture**	**Single Culture** **(at 5 × 10^5^ CFU/mL)**	**Co-Culture**
5 × 10^4^	0.394 ± 0.047 ^a^	0.324 ± 0.038 ^a^	0.249 ± 0.054 ^a^	0.230 ± 0.026 ^a^
5 × 10^5^	0.283 ± 0.031 ^a^	0.294 ± 0.031 ^a^	0.249 ± 0.054 ^a^	0.196 ± 0.034 ^a^
5 × 10^6^	0.277 ± 0.056 ^a^	0.238 ± 0.074 ^a^	0.249 ± 0.054 ^a^	0.182 ± 0.054 ^b^
(**b**) Table of population maximal (CFU/mL)				
	***Hanseniaspora uvarum* 3137**		***Metschnikowia pulcherrima* MCR24**	
**Initial Population of *Hu* (CFU/mL)**	**Single Culture**	**Co-Culture**	**Single Culture** **(at 5 × 10^5^ CFU/mL)**	**Co-Culture**
5 × 10^4^	2.27 × 10^7^ ± 0.48 × 10^7 a^	1.68 × 10^7^ ± 0.80×10^7 a^	4.58 × 10^7^ ± 1.94 × 10^7 a^	9.36 × 10^6^ ± 3.01 × 10^6 b^
5 × 10^5^	3.56 × 10^7^ ± 0.56 × 10^7 a^	3.90 × 10^7^ ± 1.29×10^7 a^	4.58 × 10^7^ ± 1.94 × 10^7 a^	5.54 × 10^6^ ± 2.02 × 10^6 b^
5 × 10^6^	4.79 × 10^7^ ± 0.81 × 10^7 a^	4.68 × 10^7^ ± 0.64×10^7 a^	4.58 × 10^7^ ± 1.94 × 10^7 a^	2.91 × 10^6^ ± 0.70 × 10^6 b^

Letters in index correspond to statistical groups (Kruskal–Wallis test α = 5%) between mono and co-cultures for each condition (comparison by species).

**Table 2 foods-13-00724-t002:** Comparison between *H. valbyensis* strain ScS and *M. pulcherrima* MCR24 in single and co-cultures for: (**a**) maximal growth rate (µmax (h^−1^)) and (**b**) maximal population (CFU/mL) reached after 72 h at 20 °C.

(**a**) Table of µmax (h^−1^)				
	***Hanseniaspora valbyensis* ScS**		***Metschnikowia pulcherrima* MCR24**	
**Initial Population of *Hv* (CFU/mL)**	**Single Culture**	**Co-Culture**	**Single Culture** **(at 5 × 10^5^ CFU/mL)**	**Co-Culture**
5 × 10^4^	0.378 ± 0.043 ^a^	0.171 ± 0.036 ^b^	0.249 ± 0.054 ^a^	0.270 ± 0.045 ^a^
5 × 10^5^	0.325 ± 0.067 ^a^	0.213 ± 0.028 ^b^	0.249 ± 0.054 ^a^	0.237 ± 0.029 ^a^
5 × 10^6^	0.209 ± 0.055 ^a^	0.177 ± 0.020 ^a^	0.249 ± 0.054 ^a^	0.213 ± 0.025 ^a^
(**b**) Table of population maximal (CFU/mL)				
	***Hanseniaspora valbyensis* ScS**		***Metschnikowia pulcherrima* MCR24**	
**Initial Population of *Hv* (CFU/mL)**	**Single Culture**	**Co-Culture**	**Single Culture** **(at 5 × 10^5^ CFU/mL)**	**Co-Culture**
5 × 10^4^	1.55 × 10^8^ ± 0.32 × 10^8 a^	9.06 × 10^6^ ± 1.43 × 10^6 b^	4.58 × 10^7^ ± 1.94 × 10^7 a^	3.34 × 10^7^ ± 2.55 × 10^7 a^
5 × 10^5^	2.72 × 10^8^ ± 0.16 × 10^8 a^	4.70 × 10^7^ ± 1.12 × 10^7 b^	4.58 × 10^7^ ± 1.94 × 10^7 a^	2.63 × 10^7^ ± 1.75 × 10^7 a^
5 × 10^6^	2.89 × 10^8^ ± 0.11 × 10^8 a^	2.33 × 10^8^ ± 0.16 × 10^8 b^	4.58 × 10^7^ ± 1.94 × 10^7 a^	1.33 × 10^7^ ± 0.29 × 10^7 b^

Letters in index correspond to statistical differences (Kruskal–Wallis test α = 5%) between mono and co-cultures for each condition (comparison by species).

**Table 3 foods-13-00724-t003:** Consumption percentages of each amino acid and ammonium in single culture after 72 h growth at 20 °C. Red bars correspond to nitrogen resources consumed by at least 50%.

	*Hanseniaspora uvarum* 3137			*Hanseniaspora valbyensis* ScS			*Metschnikowia pulcherrima*
	5 × 10^4^ CFU/mL	5 × 10^5^ CFU/mL	5 × 10^6^ CFU/mL	5 × 10^4^ CFU/mL	5 × 10^5^ CFU/mL	5 × 10^6^ CFU/mL	5 × 10^5^ CFU/mL
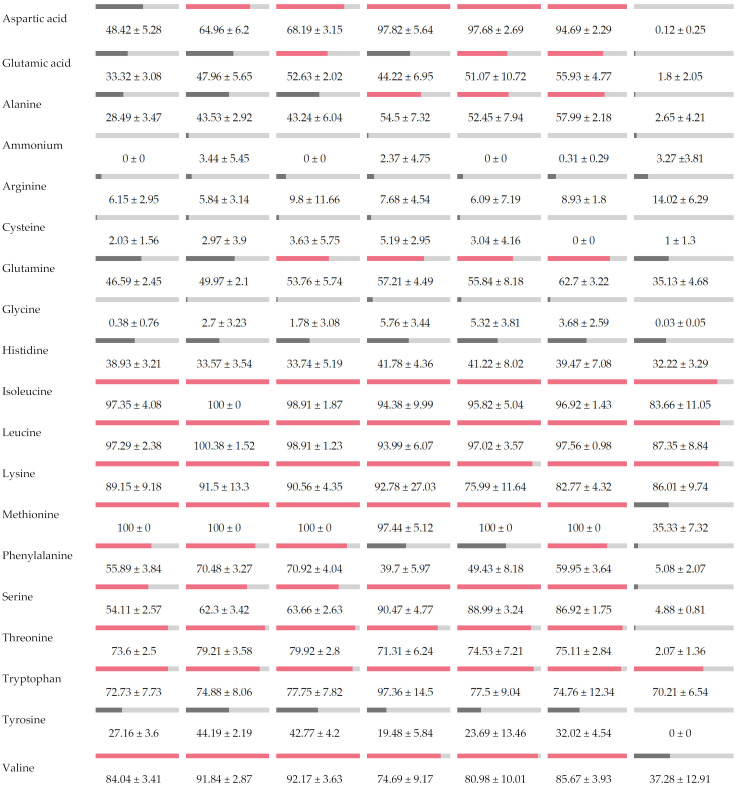							

**Table 4 foods-13-00724-t004:** Consumption parameters of preferential amino acids in single cultures at 20 °C.

	*H. uvarum*												*H. valbyensis*												*M. pulcherrima*			
	5 × 10^4^ CFU/mL				5 × 10^5^ CFU/mL				5 × 10^6^ CFU/mL				5 × 10^4^ CFU/mL				5 × 10^5^ CFU/mL				5 × 10^6^ CFU/mL				5 × 10^5^ CFU/mL			
	%24 h ^1^	Lag ^2^	Qs ^3^	Rs ^4^	%24 h	Lag	Qs	Rs	%24 h	Lag	Qs	Rs	%24 h	Lag	Qs	Rs	%24 h	Lag	Qs	Rs	24%h	Lag	Qs	Rs	%24 h	Lag	Qs	Rs
Ile	45	7.00	0.088	0.107	74	4.00	0.059	0.942	81	0.50	0.040	0.872	6	16.50	0.299	0.061	18	9.75	0.059	0.141	36	6.25	0.007	0.451	24	7.00	0.037	0.405
Leu	41	7.00	0.143	0.399	68	3.50	0.094	1.304	76	0.75	0.059	1.291	6	17.50	0.589	0.097	18	10.25	0.050	0.117	36	5.50	0.015	0.599	28	5.50	0.061	0.419
Lys	90	7.00	0.039	0.379	93	0.75	0.039	0.376	90	0.00	0.042	0.968	30	10.75	0.377	0.059	70	8.00	0.023	0.127	87	3.00	0.011	0.259	50	3.25	0.039	0.181
Trp	42	5.00	0.769	1.031	57	1.00	0.463	4.339	70	0.50	0.225	5.536	19	13.00	5.359	1.547	20	7.75	0.666	1.448	38	1.25	0.134	2.684	39	2.75	0.760	4.617

The greater the color intensity, the higher the consumption. ^1^ %24 h: Percentage of amino acid consumed in 24 h. ^2^ Lag: Time before clear start of consumption of the amino acids by yeasts (h). ^3^ Qs: Maximal speed rate of amino acids consumption by cell (ng/L/h/cell). ^4^ Rs: Maximal speed rate of amino acids consumption (ng/L/h).

**Table 5 foods-13-00724-t005:** Oxygen consumption parameters for each condition in single culture at 20 °C.

Species	Initial Concentration (CFU/mL) ^1^	Maximal Speed (mg/L/h) ^2^	T50 (h) ^3^	Total Consumption (h) ^4^
*Hanseniaspora uvarum* 3137	5 × 10^4^	1.90 ± 0.30 ^A^	8.56 ± 0.25 ^A^	11.14 ± 1.18 ^A^
	5 × 10^5^	4.35 ± 0.98 ^B^	2.33 ± 0.11 ^B^	3.78 ± 0.54 ^B^
	5 × 10^6^	7.41 ± 0.77 ^C^	1.06 ± 0.05 ^C^	2.24 ± 0.28 ^C^
*Hanseniaspora valbyensis* ScS	5 × 10^4^	2.31 ± 0.13 ^a^	9.50 ± 0.00 ^a^	11.00 ± 0.00 ^a^
	5 × 10^5^	2.90 ± 0.47 ^b^	6.33 ± 0.38 ^b^	8.03 ± 0.21 ^b^
	5 × 10^6^	2.99 ± 0.30 ^b^	2.31 ± 0.24 ^c^	4.50 ± 1.00 ^c^
*Metschnikowia pulcherrima* MCR24	5 × 10^5^	4.37 ± 0.64 ^B c^	1.93 ± 0.10 ^d D^	3.16 ± 0.19 ^d D^

Letters corresponds to statistical groups (Kruskal–Wallis test, α = 0.05), capital letters correspond to comparisons between *M. pulcherrima* and *H. uvarum* and lower-case letters to comparisons between *M. pulcherrima* and *H. valbyensis.*
^1^ Initial concentration of each strain, ^2^ Maximal speed measured, ^3^ Time to consume 50% of the total dissolved oxygen of the medium, ^4^ Time to consume all the dissolved oxygen of the medium.

## Data Availability

The original contributions presented in the study are included in the article, further inquiries can be directed to the corresponding author.

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
