# Peer review of "Competition for Nitrogen Resources: An Explanation of the Effects of a Bioprotective Strain Metschnikowia pulcherrima on the Growth of Hanseniaspora Genus in Oenology"

_foods, 2024, doi:10.3390/foods13050724_

Round 1
Reviewer 1 Report
Comments and Suggestions for Authors
This paper is about how some yeasts might to be used as biocontrol in the winemaking. It is an interesting theme.
This reviewer want to know what is te resistance of SO2 of both Hansianosporas species and Metschnikowia pulcherrima strain studied.
Other considerations:
Line 152. It is necessary indicated the ethanol concentration in the text.
Line 173. what is the biological explication to use different culture media?
What is the biological explication about of the use the described medium for Hansienospora strains?
Table1 : The values of M. pulcherrima in single culture are similar with concentrations different of Hanseniaspora 3137?
Line 321-322. Indicated in the text "Table 2".
Tables in general. Why the value have comma and not point?
Author Response
First of all, we would like to thanks you for the reviewing and for your comments and feedback.
We haven't investigated the level of resistance of SO2 of our strains, however, with an SO2 addition made during winemaking (Puyo et al 2023, Use of Oenological Tannins to Protect the Colour of Rosé Wine in a Bioprotection Strategy with Metschnikowia pulcherrima), the SO2 addition have not impacted the growth of the bioprotection.
Line 152 : The synesthetic medium described in this sentence was a synthetic must and not a synthetic wine. By consequences we don’t have any ethanol concentration to indicate.
Line 173: WL is a non-selective medium were all strain can grow, whereas ITV is a selective medium where Hanseniaspora was able to grow but not Metschnikowia. We have decided to use both medium because we assumed that Hanseniaspora genre will decrease in the medium (especially in the case where Hanseniaspora inoculation was 1 log lower that Metschnikowia concentration), so we used another medium on order to be able to enumerate Hanseniaspora strain even if their population will decrease.
Table 1: In the single culture column, the values of Metschnikowia are similar because they correspond to the single culture growth of Metschnikowia, which was used as the control condition for Metschnikowia growth parameters.
Comma in tables have been replaced.
Hoping the changes will meet your expectations.
Reviewer 2 Report
Comments and Suggestions for Authors
The manuscript “Competition for nitrogen resources: an explanation of the effects of a bioprotective strain Metschnikowia pulcherrima on the growth of Hanseniaspora genus in oenology” reports results about the possible bio-protection mechanisms of Metschnikowia pulcherrima against two different species of Hanseniaspora (competition for nitrogenous resources, in this case). Bioprotection is a biological alternative to SO2 in winemaking to prevent grape musts microbial alteration. However, at practical level, the use of bioprotection in wine making not always give the expected result. So, the results exposed in this study are of interest to the scientific community and winemakers. Therefore, I recommend minor changes.
Comments and suggestions:
Key words: You could include the non -saccharomyces species used in the study
Line 86-90: Revise this sentence
Line 123: “contamination” I think is better “concentration”
Mat met
Line 137: glycerol 20%. Is this correct?
Supplementary material: There are two Tables S1, please revise it
Results and discussion:
Line 277. How Maximal growth rate (μmax) was calculated. Please, indicated it in material and methods or in table foot
Table 1. Include Full name of M. pulcherrima
Line 450: Roca -mesa et al , 2020 (include ref. number-43)
Paragraph 587-599: Could you indicate what CHO, CHON (aromatic compounds?), CHONS (sulfur-containing peptides?), and CHOS chemical families/groups represent?
Line 604: Han et al., 2020; Prothstein and Hart, 1964 (Ref. numbers 68 and 69)
Figure 4. What does the pie chart in the graphics mean?
Include the list on supplementary materials
References:
Please, revise the journal style and check the name of the species in titles
Author Response
First of all, we would like to thanks you for the reviewing and for yours comments and feedback.
We added the name of NS species used as key words.
Line 137: Yes it was stored in a solution of glycerol at 20%, we added the information 20% (v/v) inside the text
We changed the tables in supplementary data.
Paragraph 587-599: precision is given on the mat & met. CHO, CHON, CHONS, CHOS compounds which are classified according to their chemical formula. This information gives us an idea of their elemental composition only.
The pie chart (figure 4) represents the putative chemical families, but we have forgotten to add the legend, we change the figure.
We have added the supplementary data list at the end of the manuscript
References have been inserted according to the Editor recommendation (With Zotero software)
Hoping the changes will meet your expectations.
Reviewer 3 Report
Comments and Suggestions for Authors
I have reviewed a paper entitled "Competition for nitrogen resources: an explanation of the effects of a bioprotective strain Metschnikowia pulcherrima on the growth of the Hanseniaspora genus" by Puyo et al. The paper is generally well-written and structured, based on comprehensive laboratory research. The experiments are well-planned, and the analyses were conducted using appropriate methods. The article is sufficiently novel. There is a thorough discussion of the results obtained, and all conclusions logically follow from the presented data and discussion.
As a result, I recommend minor revisions. Further comments:
- Please correct the notation of CFU/mL values throughout the manuscript (in the current notation, it looks like, e.g., 5.10^5, not 5 * 10^5).
- Figures 1 and 2 should be placed in the 3.1.1 section, not at the end of the manuscript.
- Data in tables 1, 2, and 4 should use English notation, with periods instead of commas.
- Whenever used, please explain the statistical groups denoted by the Kruskal-Wallis test.
- Line 326 - the reference to Table 3b should be changed to 2b.
- Please standardize the notation of units for μmax (1/h) so that it is consistent with the notation for the others.
- Table 4 (page 12) - Qs and Rs should be changed to "Maximal speed rate of amino acids consumption"
- Figure 4 (panels b-d, page 19) - please explain what the pie charts, % values, and different colours mean.
Supplementary material:
- Figure S1 - for clarity, I recommend a colour change in panels a and b.
- Figure S2 - lacks a more detailed description, especially regarding the groups of biomarkers shown.
Author Response
First of all, we would like to thanks you for the reviewing and for yours comments and feedback.
The notation “5.105” was the notation allowed by the template of Foods.
On the MDPI templates there is a special section after the results to put figures. That’s why we placed figures on this section.
The explanation of the statistical groups are given in tables footnotes. Table 5 : Letters corresponds to statistical groups (Kruskal Wallis test, a = 0.05), capital letters correspond to comparisons between M. pulcherrima and H. uvarum and lower-case letters to comparisons between M. pulcherrima and H. valbyensis. And table 1 and 2: Letters in index correspond to statistical differences (Kruskal-Wallis test α = 5%) between mono and co-cultures for each condition (comparison by species).
Line 326 : Table 3b have been changes in 2b.
Table 4 : The word “consumption” have been added in the table legend
Figure 4 : Wa have added the figure legend
Figure S2 : The heat map was made according to the biomarkers extracted for each condition. The biomarkers details (their chemical composition) are given inside the figure 4, we referred to Figure 4 in the legend of the figure S2.
Hoping the changes will meet your expectations.
Round 2
Reviewer 1 Report
Comments and Suggestions for Authors
Thanks for the authors for yours answers, but there two questions that they don´t respond
1) In the case of SO2 is interesting its analysis because is important that biocontrol yeast have lower resistence than S.cervisiae or other spoilage yeast, then the use of SO2 will be minor in winemaking if is used the biocontrol yeast.
2) The authors indicate: Line 173: "WL is a non-selective medium were all strain can grow, whereas ITV is a selective medium where Hanseniaspora was able to grow but not Metschnikowia".
What is the possible biological explication?
Author Response
Dear reviewer,
Concerning your first request, as mentioned previously, we have not investigated the level of SO2 resistance of our strains, and it would be interesting to investigate this parameter in the future. However, the use of bioprotection aimed to decrease the amount of SO2 added during the pre-fermentative steps of winemaking, so the lower SO2 concentration applied during winemaking in a bioprotective strategy is not because of the lower SO2 resistance of the bioprotective yeast but thanks to the ability of the bioprotective yeast to microbiologically protects the must, leading to a lower SO2 level needed.
Concerning your second question, WL is a non-selective medium because it was made with all the classical nutrients required for the yeasts’ growth, while on ITV medium a selective compound was added (actidione) which inhibits the growth of M. pulcherrima but not the growth of both Hanseniaspora strains which are resistant to this compound.
Hopping our answer will help to clarify.